# Heart Rate-Dependent Degree of Motion Artifacts in Coronary CT Angiography Acquired by a Novel Purpose-Built Cardiac CT Scanner

**DOI:** 10.3390/jcm11154336

**Published:** 2022-07-26

**Authors:** Milán Vecsey-Nagy, Ádám Levente Jermendy, Márton Kolossváry, Borbála Vattay, Melinda Boussoussou, Ferenc Imre Suhai, Alexisz Panajotu, Judit Csőre, Sarolta Borzsák, Daniele Mariastefano Fontanini, Csaba Csobay-Novák, Béla Merkely, Pál Maurovich-Horvat, Bálint Szilveszter

**Affiliations:** 1MTA-SE Cardiovascular Imaging Research Group, Heart and Vascular Center, Semmelweis University, 1122 Budapest, Hungary; adam.jermendy@gmail.com (Á.L.J.); martonandko@gmail.com (M.K.); bori.vattay@gmail.com (B.V.); melinda.b.md@gmail.com (M.B.); suhaiimi987@gmail.com (F.I.S.); panajotualexisz@gmail.com (A.P.); sati.borzsak@gmail.com (S.B.); maurovich.horvat@gmail.com (P.M.-H.); balint.szilveszter@gmail.com (B.S.); 2Heart and Vascular Center, Semmelweis University, 1122 Budapest, Hungary; csorejudit@gmail.com (J.C.); fontanini.med@gmail.com (D.M.F.); csaba@csobay.hu (C.C.-N.); szivct@gmail.com (B.M.); 3Medical Imaging Centre, Semmelweis University, 1082 Budapest, Hungary

**Keywords:** coronary artery disease, computed tomography angiography, heart rate, artifacts

## Abstract

Although reaching target heart rate (HR) before coronary CT angiography (CCTA) is still of importance, adequate HR control remains a challenge for many patients. Purpose-built cardiac scanners may provide optimal image quality at higher HRs by further improving temporal resolution. We aimed to compare the amount of motion artifacts on CCTA acquired using a dedicated cardiac CT (DCCT) compared to a conventional multidetector CT (MDCT) scanner. We compared 80 DCCT images to 80 MDCT scans matched by sex, age, HR, and coronary dominance. Image quality was graded on a per-patient, per-vessel and per-segment basis. Motion artifacts were assessed using Likert scores (1: non-diagnostic, 2: severe artifacts, 3: mild artifacts, 4: no artifacts). Patients were stratified into four groups according to HR (<60/min, 60–65/min, 66–70/min and >70/min). Overall, 2328 coronary segments were evaluated. DCCT demonstrated superior overall image quality compared to MDCT (3.7 ± 0.4 vs. 3.3 ± 0.7, *p* < 0.001). DCCT images yielded higher Likert scores in all HR ranges, which was statistically significant in the 60–65/min, 66–70/min and >70/min ranges (3.9 ± 0.2 vs. 3.7 ± 0.2, *p* = 0.008; 3.5 ± 0.5 vs. 3.1 ± 0.6, *p* = 0.048 and 3.5 ± 0.4 vs. 2.7 ± 0.7, *p* < 0.001, respectively). Using a dedicated cardiac scanner results in fewer motion artifacts, which may allow optimal image quality even in cases of high HRs.

## 1. Introduction

Coronary CT angiography (CCTA) has emerged as a robust non-invasive diagnostic tool for the assessment of coronary artery disease (CAD) and has recently received a class I recommendation as the initial test for symptomatic patients in whom obstructive coronary stenosis cannot be excluded clinically [1,2,3]. The limited temporal resolution of conventional multidetector CT (MDCT) scanners, however, may compromise the diagnostic performance of CCTA by producing substantial motion artifacts [4,5]. Although optimal heart rate (HR) control is still needed to achieve adequate image quality, the presence of motion artifacts still poses a technical challenge that potentially hinders the accurate assessment of the coronary arteries, and given that up to one-fourth of patients with suboptimal HR are non-responders to pre-scan beta-blockade, novel technical advancements reducing motion artifacts may potentially provide improved care for a wide array of challenging patients [6,7].

The technical parameters of the world’s first purpose-built, dedicated cardiac CT (DCCT) scanner (CardioGraphe, GE Healthcare, Chicago, IL, USA) were optimized to permit visualization of the entire heart and coronary system in a single heartbeat. This purpose-built cardiac CT applies two overlapping cone beams from two X-ray sources spread along the Z-axis, providing a coverage of 14 cm that allows the imaging of the entire coronary system in a single heartbeat (as compared to the 8 cm coverage of MDCT), precluding step artifacts. Furthermore, the focused field of view (25 cm) provided by the scanner yields state-of-the-art spatial resolution [8]. A further important technological advancement was the substantial acceleration of gantry rotation time, making DCCT’s 240 msec rotation speed one of the fastest commercially available solutions [9].

While these technical advancements may prove beneficial for eliminating motion artifacts during coronary imaging, detailed evaluation of DCCT’s performance upon different HR ranges is yet to be investigated. Our aim was to evaluate the effect of DCCT compared with MDCT on per-patient, per-vessel and per-segment image quality parameters and interpretability in CCTA datasets.

## 2. Materials and Methods

### 2.1. Study Population

Overall, CCTA images of 160 patients referred to clinically indicated CCTA were analyzed in a tertiary referral center. Patients older than 18 years who all gave approval to data retrieval and analysis were included. Cases where substantial breathing artifacts were present were excluded from the current study. In addition, we excluded patients with extrasystole or arrhythmia during the acquisition of the scan. Demographic parameters and clinical data were collected anonymously.

Overall, 80 DCCT scans were enrolled prospectively between the period of September 2019 and March 2020, equally divided between the four predetermined HR ranges (<60 beats/min [bpm], 60–65 bpm, 66–70 bpm, >70 bpm). The control group of 80 MDCT patients were selected from our institutional cardiac CT registry (Axis, Neuman Medical Ltd., Budapest, Hungary) containing 9090 previously performed CCTA datasets and structured reports. The control cases were selected according to the following matching criteria: sex, age, HR, acquisition phase (systole/diastole) and coronary dominance. Regarding age, a tolerance of ±10% was applied, whereas a maximum difference of ±2 bpm was allowed for HR (Figure 1). Finally, subgroups of patients were created by assigning each patient to one of the following predefined HR ranges: <60 bpm, 60–65 bpm, 66–70 bpm, >70 bpm. Sample size calculations were conducted based on 758 DCCT and 1518 MDCT scans from the previous year. Based on the observed differences in overall image quality (3.2 ± 0.4 vs. 2.7 ± 0.4, respectively), we calculated that a sample size of 20 would be needed in each HR category (with 80% power and a 2-sided alpha level of 0.05).

### 2.2. CT Acquisition

CCTA examinations were performed using a conventional MDCT (Brilliance iCT 256, Philips Healthcare, Best, The Netherlands) or a DCCT scanner (CardioGraphe, GE Healthcare, Chicago, IL, USA) with prospective ECG-triggered axial acquisition mode. Key specifications and acquisition protocols for the two scanners are listed in Table A1 in Appendix A. Tube voltage and tube current were adjusted according to patient size. Image acquisition was performed with 270 and 240 msec rotation time for MDCT and DCCT, respectively. If the patient’s HR exceeded 65 beats/min, a maximum of 20 mg metoprolol was given intravenously for heart rate control under the supervision of a physician (Table A2). All patients received sublingual nitroglycerin (0.8 mg) to induce proper vasodilation during CCTA. Systolic triggering was applied when HR of the patient was above 80 bpm. Iomeprol contrast material (Iomeron 400, Bracco Ltd., Milan, Italy) through antecubital venous access was used with 85–95 mL contrast agent at a flow rate of 4.5–5.5 mL/sec using a four-phasic protocol [10]. In order to obtain proper scan timing, bolus tracking in the left atrium was used.

The radiation dose was recorded by the machine as the dose-length product, which was converted to the effective dose in mSv by multiplying by the conversion factor of 0.014 mSv × mGy^−1^ × cm^−1^ [11].

### 2.3. CTA Image Reconstruction

Iterative reconstructions from the original data were implemented on both scanner terminals during postprocessing. The raw data were retrospectively reprocessed using iDose5 (Philips Healthcare, Best, Netherlands) for MDCT and ASiR (GE Healthcare, Chicago, IL, USA) for DCCT. All datasets were reconstructed with a standard kernel routinely applied in clinical practice (MDCT: XCC, DCCT: CV standard).

### 2.4. Assessment of Image Quality

All datasets were anonymized and then reviewed and interpreted on a commercially available DICOM viewer (RadiAnt, version 2.3; Medixant, Poznan, Poland). Evaluation of the images was performed by two experienced readers (a cardiologist with 7 years and a radiologist with 9 years of experience in cardiac CT) blinded for the type of scanner used. All coronary segments with a diameter above 1.5 mm were assessed using the 18-segment model of the Society of Cardiovascular CT (SCCT) [12]. According to the SCCT model, we considered segments 1 and 2; segments 5, 6 and 7; and segments 11, 12 and 16 as proximal right coronary artery (RCA), proximal left anterior descending artery (LAD) and proximal left circumflex artery (LCx). A four-point Likert scale was applied to rate subjective image quality on axial slices and multiplanar reformations (Figure 2). In case of initial discordance between the readers, a consensus was achieved during a joint reading. The readers were instructed to ignore issues that could not be ascribed to the presence of motion artifacts (e.g., prominent image noise, poor contrast, extensive calcification, or step artifacts). Subjective image quality was evaluated on the best phases selected by the examiners with regards to the degree of motion artifacts on a per-segment level: non-diagnostic, with severe motion artifacts impairing accurate evaluation (1); moderate, with considerable motion artifacts, only sufficient to rule out significant luminal stenosis (2); good, with preserved ability to assess the degree of luminal stenosis (3) and excellent, with no visible motion artifacts (4). Interpretability was defined on a per-patient, per-coronary and per-segment basis: if an evaluated coronary segment was rated as non-diagnostic, the corresponding artery and patient was considered non-interpretable.

### 2.5. Statistical Analysis

Continuous variables are expressed as medians and interquartile ranges (IQRs) and categorical variables as numbers and percentages. The Kolmogorov–Smirnov test was applied to evaluate the normality of continuous parameters. Continuous data was compared between the MDCT and DCCT groups using a paired samples *t*-test, while a chi-square test was used to assess differences between categorical variables. Per-patient, per-coronary and per-segment comparison of image quality scores was performed by a Wilcoxon matched pairs signed-rank test for the entire cohort and for each of the four HR groups, as well. To determine interobserver agreement, the intra-class correlation coefficient was calculated for Likert scores, while Cohen’s kappa was measured as an indicator of reproducibility for interpretability. A two-sided *p* < 0.05 was considered significant in all analyses. SPSS (Armonk, NY, USA, version 27.0) was used for all calculations.

## 3. Results

### 3.1. Demographic and Scanning Parameters

In total, 160 patients were included in our study, equally divided between the conventional MDCT and DCCT scanners. The mean age of the patients was 59.4 ± 10.2 and 58.9 ± 10.2 years, respectively. No statistically significant difference could be observed between the two groups regarding anthropometric data and cardiovascular risk factors. Mean dose-length product [359.1 ± 83.6 vs. 294.6 ± 114.6 mGy*cm (*p* < 0.001)] and effective dose [5.0 ± 1.2 vs. 4.1 ± 1.6 mGy*cm (*p* < 0.001)] were significantly lower in the DCCT group. Patient demographics and CT acquisition characteristics are provided in Table 1.

### 3.2. Distribution of Likert Scores

A total of 2328 segments of 480 coronaries were analyzed, 1019 segments in the MDCT and 989 segments in the DCCT group. The distribution of the Likert scores in the two subgroups is displayed in Figure 3. In the DCCT group, the proportion of segments with excellent image quality significantly exceeded that in the MDCT population (743/989 (75.1%) vs. 572/1019 (56.1%) (*p* < 0.001)), whereas the number of non-diagnostic DCCT segments was substantially lower as compared to the MDCT group (11/989 (1.2%) vs. 85/1019 (8.4%) (*p* < 0.001)) (Figure 4).

Overall, 16 datasets of the 160 were reevaluated by a third reader (radiology trainee with three years of experience in CCTA) to assess interreader variability. A good agreement was found between the readers for image quality, with an ICC of 0.81 regarding Likert scores and a Cohen’s kappa value of 0.71 for interpretability.

### 3.3. Image Interpretability

The use of DCCT resulted in a substantial improvement in interpretability (segments were deemed interpretable with Likert score >1) on per-patient (74/80 (92.5%) vs. 52/80 (65.0%) (8.4%) (*p* < 0.001)), per-coronary (199/240 (82.9%) vs. 232/240 (96.7%) (*p* < 0.001)) and per-segment (934/1019 (91.7%) vs. 978/989 (98.9%) (*p* < 0.001)) levels, with a significant reduction in the number of non-diagnostic segments. Regarding the three major epicardial coronaries, similar differences could be observed for LM-LAD (69/80 (86.3%) vs. 79/80 (98.8%) (*p* = 0.003)), LCx (66/80 (82.5%) vs. 78/80 (97.5%) (*p* = 0.002)) and RCA (60/80 (75.0%) vs. 75/80 (93.8%) (*p* = 0.002)). Detailed assessment of interpretability is displayed in Table 2.

### 3.4. Subjective Image Quality on Per-Patient, Per-Vessel and Per-Segment Levels

In our entire cohort of 160 patients, DCCT yielded significantly better overall image quality than conventional MDCT on a per-patient, per-coronary and per-segment level, as well. The final visual score on a per-patient basis was 3.7 ± 0.4 vs. 3.3 ± 0.7 (*p* < 0.001). Likert scores of the major coronaries also differed significantly between the scanners (LM-LAD: 3.8 ± 0.3 vs. 3.5 ± 0.6 (*p* < 0.001), LCx: 3.8 ± 0.5 vs. 3.3 ± 0.9 (*p* < 0.001), RCA: 3.5 ± 0.6 vs. 3.0 ± 0.9 (*p* < 0.001)). Similarly, DCCT performed better in both the proximal (3.7 ± 0.3 vs. 3.4 ± 0.6 (*p* < 0.001)) and distal (3.6 ± 0.5 vs. 3.2 ± 0.8 (*p* < 0.001)) coronary segments, with more prominent differences observed in the proximal slices (Table 3).

Although Likert scores of DCCT scans displayed a consistent tendency of superiority compared with MDCT studies, no significant differences could be noted between the groups in our subcohort of patients with an HR below 60 bpm.

Regarding patients with a bpm between 60 and 65, image quality score on a per-patient basis using DCCT was significantly higher than that with MDCT (3.9 ± 0.2 vs. 3.7 ± 0.2 (*p* = 0.008)). The most marked differences in this HR group pertained to RCA (3.8 ± 0.4 vs. 3.4 ± 0.4 (*p* = 0.003)); however, proximal segments also displayed greater mean scores using DCCT (3.9 ± 0.2 vs. 3.7 ± 0.3 (*p* = 0.01)).

Per-patient subanalysis of participants with a bpm between 66 and 70 demonstrated significantly better Likert scores for DCCT than MDCT (3.5 ± 0.5 vs. 3.2 ± 0.6 (*p* = 0.048)). Furthermore, LM-LAD performed significantly better for DCCT in this heart range (3.7 ± 0.4 vs. 3.4 ± 0.6 (*p* = 0.04)).

Patients with an HR above 70 bpm achieved a higher overall mean score for image quality (3.5 ± 0.4 vs. 2.7 ± 0.7 (*p* = 0.04)). Using DCCT, LM-LAD (3.7 ± 0.3 vs. 3.0 ± 0.6 (*p* = 0.02)), LCx (3.6 ± 0.5 vs. 2.5 ± 1.0 (*p* = 0.001)) and RCA (3.3 ± 0.5 vs. 2.3 ± 0.9 (*p* = 0.003)) all demonstrated better image quality than MDCT. DCCT proved to be superior in a per-segment analysis, as well, which was reflected in the Likert scores of both the proximal (3.6 ± 0.3 vs. 2.8 ± 0.7 (*p* < 0.001)) and distal (3.5 ± 0.5 vs. 2.5 ± 0.9 (*p* = 0.002)) segments.

## 4. Discussion

The main finding of our study is that CCTA performed with a new-generation DCCT permits the visualization of coronaries with superior image quality compared to a conventional MDCT in all HR ranges above 60 bpm with a reasonably low radiation exposure. Although the rate of interpretability and image quality scores were higher in all HR ranges for DCCT, the most pronounced differences could be observed above 70 bpm.

The endorsement of CCTA as an initial non-invasive test for those symptomatic patients in whom obstructive CAD cannot be excluded clinically has recently been integrated to the recommendations of the European Society of Cardiology [3]. The initiative falls in line with previous recommendations of the National Institute of Health and Care Excellence in the UK, where CCTA was recommended as the initial diagnostic test for patients with stable chest pain [13]. In correspondence with these guidelines, a substantial increase of near 700% is estimated in CCTA delivery across the UK [14]. Consequently, technical innovations improving the diagnostic ability of CCTA are warranted in order to provide appropriate care to a gradually increasing number of patients.

Significant advances have been made in the past years regarding CT hardware, including faster gantry rotation speeds, increased spatial resolution, and detector coverage [15]. Despite continuous improvements in spatial and temporal resolution, however, up to 10% of visualized coronary segments are still judged as non-diagnostic on CCTA [16]. A major source of non-interpretable segments on CCTA datasets are motion artifacts, which predominantly occur when the temporal resolution exceeds the motion-free interval of the given coronary segments [2,17]. According to previous investigations, the mid RCA, mid LAD and distal LAD, in particular, are susceptible to motion artifacts at high HRs [18]. In our present study, apart from their significant superiority at >70 bpm, RCA and LAD visualized by DCCT performed better than by MDCT in the 60–65 bpm and 66–70 bpm ranges, respectively. Current guidelines recommend reaching a target HR of <60 bpm by administering oral and/or intravenous medication [19]. Since in approximately half of the patients the target heart rate cannot be achieved [20], it seems plausible that the use of DCCT could be beneficial for a broad range of patients.

A further consideration that should be taken into account is that approximately one-fourth of patients do not demonstrate adequate response to HR control [6,7]. Furthermore, approximately 5–11% of patients have contraindications or intolerance to the most frequently used HR-lowering medications [21,22]. Our results provide evidence that DCCT can be advantageous for non-responder patients or subjects with contraindication to HR-lowering medication, as DCCT proved to be substantially superior to conventional MDCT HRs above 70 bpm.

Both software- and hardware-based solutions have been developed to potentiate the reduction of motion artifacts. Software-based innovations for motion correction, such as the vendor-specific SnapShot Freeze (GE Healthcare, Chicago, IL, USA) algorithm, offer cost-effective, generally applicable alternatives that improve image quality and interpretability even for patients with insufficient HR control [17,23,24]. As the effect of software-based solutions was not the primary focus of the current investigation, images acquired using SnapShot Freeze were not used. On the other hand, current high-end CT systems continue to expand the physical limits of hardware, especially in detector, tube, and gantry technology. The world’s first dedicated cardiac CT scanner was installed in 2018 with the intention of serving the increasing demand for CCTA, and early clinical evidence supports its robustness and clinical utility [8]. The scanner provides whole heart imaging in a single heartbeat by utilizing two overlapping cone beams from two X-ray sources spread across the Z-axis. Due to its small footprint (5 m^2^), the scanner operates with a focused field of view (user-selected 160 mm or 250 mm). Combined with an ultrafast gantry rotation speed of 0.24 s, the temporal resolution offered by DCCT is 120 msec. These novel technological advancements may potentiate the reduction of motion artifacts as well as the radiation dose exposure associated with CCTA. This notion was supported in our cohort of 160 patients, as the DCCT group was exposed to a significantly lower radiation dose, as compared to conventional MDCT.

The current study has limitations that should be considered: first, the subjective nature of the image quality scoring introduces the possibility of bias into the study. It should be emphasized that although readers were blinded to information pertaining to the CT acquisition, it is, nevertheless, impossible to exclude the possibility that the type of scanner could be deduced from the datasets. Second, diagnostic accuracy was not assessed by correlating our results to invasive coronary angiography, although this was not the primary aim of our study. Although only one dedicated cardiac scanner was commercially available at the time of the current study, it may be considered a limitation that only one type of cardiac scanner was assessed. Finally, although we propose the potential utility of DCCT in radiation dose reduction, our cohorts were not matched with regards to BMI; thus, the comparison of dose performance is of limited value.

## 5. Conclusions

In conclusion, our results revealed that the utilization of this novel purpose-built cardiac CT scanner for patients undergoing CCTA results in significant improvements in image quality and interpretability, with the most beneficial impact at higher HR. With further studies supporting its favorable radiation dose performance, the implementation of this dedicated CT in clinical routine is expected to provide access to a wider range of challenging patients.

## Figures and Tables

**Figure 1 jcm-11-04336-f001:**
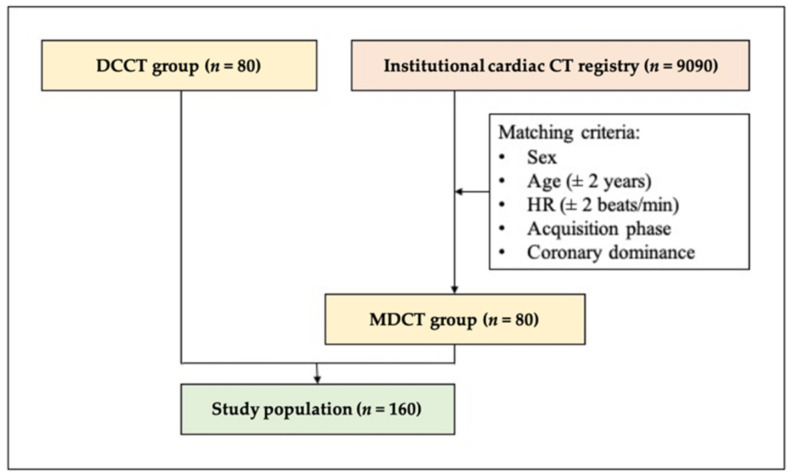
Flowchart describing the selection of the study cohort. DCCT, dedicated cardiac CT; HR, heart rate; MDCT, multidetector CT. All patients provided written informed consent prior to the examination. The study was approved by the Scientific and Research Ethics Committee of the Hungarian Medical Research Council and was carried out in accordance with the tenets of the Declaration of Helsinki.

**Figure 2 jcm-11-04336-f002:**
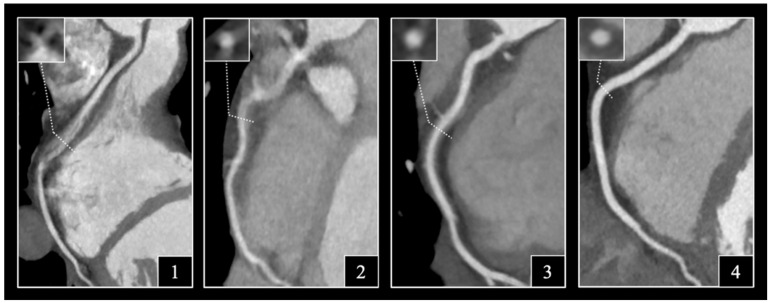
Representative curved multiplanar reconstructions and corresponding cross-sectional images demonstrating examples of the 4-point Likert scale describing motion artifacts: (1), non-diagnostic image quality precluding the evaluation of the right coronary artery; (2), moderate image quality merely allowing the exclusion of obstructive stenosis; (3), good image quality with minor artifacts and (4), excellent image quality with no artifacts present.

**Figure 3 jcm-11-04336-f003:**
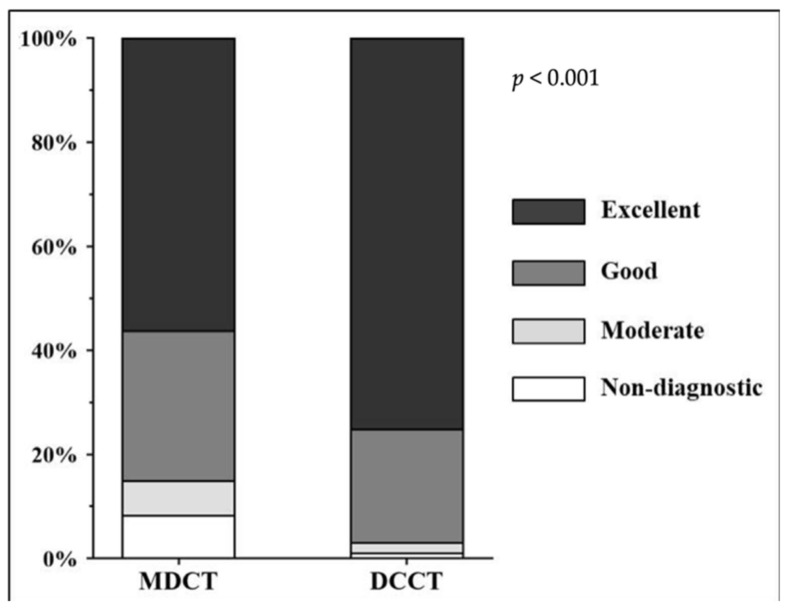
Proportion of coronary segments with excellent, good, moderate and non-diagnostic image quality in the MDCT and DCCT subgroups. MDCT, multidetector CT; DCCT, dedicated cardiac CT.

**Figure 4 jcm-11-04336-f004:**
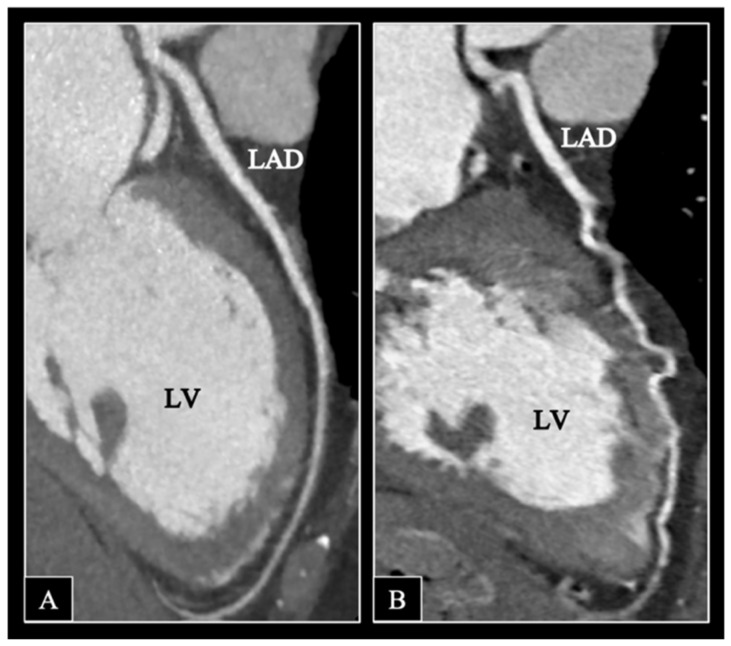
Curved multiplanar reconstruction images of the left anterior descending artery (LAD) of a patient visualized using DCCT (**A**) and a matched MDCT subject (**B**). (**A**). 61-year-old female patient with an average heart rate of 68 bpm during coronary CT angiography. LAD is depicted with no visible motion artifacts present (Likert score: 4). (**B**). Substantial motion artifacts limit the interpretability of the LAD in a 59-year-old female patient with a bpm of 67/min (Likert score: 2). DCCT, dedicated cardiac CT; MDCT, multidetector CT; bpm, beats/min; LV, left ventricle. LAD, anterior descending; LCX, left circumflex; RCA, right coronary artery.

**Table 1 jcm-11-04336-t001:** Patient demographics and imaging parameters.

	DCCT (*n* = 80)	MDCT (*n* = 80)	*p*
Demographics			
Age (years)	60.0 (50.9–66.6)	62.5 (50.3–67.7)	0.08
Female sex, *n* (%)	32 (40.0)	32 (40.0)	1.00
BSA (m^2^)	2.0 (1.8–2.1)	2.0 (1.8–2.2)	0.24
BMI (kg/m^2^)	27.4 (25.1–30.1)	27.8 (25.3–31.4)	0.22
Cardiovascular risk factors			
Current smoker, *n* (%)	15 (18.8)	17 (21.2)	0.69
Hypertension, *n* (%)	49 (61.3)	53 (66.3)	0.50
Diabetes mellitus, *n* (%)	8 (10.0)	8 (10.0)	1.00
Dyslipidemia, *n* (%)	43 (53.8)	43 (53.8)	1.00
CTA characteristics			
Diastolic triggering, *n* (%)	73 (91.3)	73 (91.3)	1.00
DLP (mGy*cm)	245.4 (243.2–343.1)	362.3 (356.7–375.9)	<0.001
Effective dose (mSv)	3.4 (3.4–4.8)	5.1 (5.0–5.3)	<0.001
Average heart rate (1/min)	65.0 (60.0–70.5)	65.0 (60.0–70.0)	0.40

Continuous variables are expressed as median and interquartile range (IQR) and categorical variables are expressed as numbers and percentages. DCCT, dedicated cardiac scanner; MDCT, multidetector scanner; BSA, body surface area; BMI, body mass index; CTA, CT angiography; DLP, dose length product.

**Table 2 jcm-11-04336-t002:** Detailed assessment of interpretability.

	DCCT	MDCT	*p*
Overall interpretability			
Per-patient	74/80 (92.5)	52/80 (65.0)	<0.001
Per-coronary	232/240 (96.7)	199/240 (82.9)	<0.001
Per-segment	978/989 (98.9)	934/1019 (91.7)	<0.001
Interpretability by coronary artery			
LM-LAD	79/80 (98.8)	69/80 (86.3)	0.003
LCX	78/80 (97.5)	66/80 (82.5)	0.002
RCA	75/80 (93.8)	60/80 (75.0)	0.002

Data are presented as *n*/N (%). All segments with a Likert score >1 were regarded as interpretable. The rate of interpretability was compared between the groups using a chi-square test. DCCT, dedicated cardiovascular CT; MDCT, multidetector CT; LM-LAD, left main-left anterior descending; LCX, left circumflex; RCA, right coronary artery.

**Table 3 jcm-11-04336-t003:** Qualitative assessment of image quality.

	DCCT	MDCT	*p*
Overall image quality			
Per-patient	3.7 ± 0.4	3.3 ± 0.7	<0.001
Per-coronary			
LM-LAD	3.8 ± 0.3	3.5 ± 0.6	<0.001
LCx	3.8 ± 0.5	3.3 ± 0.9	<0.001
RCA	3.5 ± 0.6	3.0 ± 0.9	<0.001
Per-segment			
Proximal segments	3.7 ± 0.3	3.4 ± 0.6	<0.001
Distal segments	3.6 ± 0.5	3.2 ± 0.8	<0.001
HR < 60/min			
Per-patient	3.9 ± 0.1	3.7 ± 0.4	0.09
Per-coronary			
LM-LAD	3.9 ± 0.3	3.9 ± 0.3	0.53
LCx	4.0 ± 0.1	3.8 ± 0.4	0.08
RCA	3.8 ± 0.3	3.5 ± 0.9	0.13
Per-segment			
Proximal segments	3.9 ± 0.1	3.7 ± 0.4	0.14
Distal segments	3.9 ± 0.2	3.7 ± 0.5	0.16
HR: 60–65/min			
Per-patient	3.9 ± 0.2	3.7 ± 0.2	0.008
Per-coronary			
LM-LAD	3.9 ± 0.1	3.9 ± 0.2	0.42
LCx	3.9 ± 0.3	3.7 ± 0.5	0.15
RCA	3.8 ± 0.4	3.4 ± 0.4	0.003
Per-segment			
Proximal segments	3.9 ± 0.2	3.7 ± 0.3	0.01
Distal segments	3.9 ± 0.2	3.7 ± 0.3	0.06
HR: 66–70/min			
Per-patient	3.5 ± 0.5	3.2 ± 0.6	0.048
Per-coronary			
LM-LAD	3.7 ± 0.4	3.4 ± 0.6	0.04
LCx	3.5 ± 0.7	3.2 ± 0.7	0.12
RCA	3.1 ± 0.6	2.9 ± 0.8	0.41
Per-segment			
Proximal segments	3.6 ± 0.4	3.3 ± 0.5	0.06
Distal segments	3.3 ± 0.7	3.0 ± 0.8	0.20
HR > 70/min			
Per-patient	3.5 ± 0.4	2.7 ± 0.7	<0.001
Per-coronary			
LM-LAD	3.7 ± 0.3	3.0 ± 0.6	0.002
LCx	3.6 ± 0.5	2.5 ± 1.0	0.001
RCA	3.3 ± 0.5	2.3 ± 0.9	0.003
Per-segment			
Proximal segments	3.6 ± 0.3	2.8 ± 0.7	<0.001
Distal segments	3.5 ± 0.5	2.5 ± 0.9	0.002

Qualitative parameters were compared between the scanners using a Wilcoxon signed-rank test. DCCT, dedicated cardiovascular CT; MDCT, multidetector CT; HR, heart rate; LM-LAD, left main-left anterior descending; LCX, left circumflex; RCA, right coronary artery.

## Data Availability

The data presented in this study are available on request from the corresponding author. The data are not publicly available due to reasons pertaining to patient privacy.

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
