# Peer review of "Heart Rate-Dependent Degree of Motion Artifacts in Coronary CT Angiography Acquired by a Novel Purpose-Built Cardiac CT Scanner"

_jcm, 2022, doi:10.3390/jcm11154336_

Round 1
Reviewer 1 Report
This is a well-designed and meticulously conducted study on the ability of a new generation DCCT to reduce the rate of motion artifacts in comparison to standard MDCT, especially when the heart rate during the scan is relatively high.
The study is neatly planned. The authors carefully matched the groups and “isolated” the target question of coronary motion by ignoring other possible causes of reduced image quality therefore focusing on this aspect only. They used well established criteria to grade coronary motion and asked for two readers to independently analyze image quality. The statistical analysis is well describred.
Acquisition and reconstruction on both scanners are well described.
The results are presented in a systematically detailed and clear manner accompanied by illustrative images.
The introduction is sufficient, and the discussion is relevant and focused on the manuscript’s topic.
I have no significant comments the address the authors
Author Response
We are extremely grateful for the Reviewer for the thorough revision and the positive reception of our work.
Reviewer 2 Report
1. Study is well done. However why is MDCT utilized when there is dedicated cardiac CCTA at the institute with improved quality and less radiation.
2. See metoprolol 20mg IV to reach the goal? how often is it given? Is it supervised by physicians or RN.
Author Response
This study is well done. However why is MDCT utilized when there is dedicated cardiac CCTA at the institute with improved quality and less radiation.
We would like to thank the Reviewer for the kind compliment. Our dedicated cardiac scanner was installed in December 2017, while our conventional CT has been in use since 2010. Given that in our inclusion period (September 2019 – March 2020) only DCCT scans were enrolled consecutively, the control MDCT scans were selected from our structured cardiac CT registry containing the dataset of more than 9000 previously performed coronary CTAs.
The Methods section was amended accordingly to clarify the retrospective nature of the matching:
‘The control group of 80 MDCT patients were selected from our institutional cardiac CT registry (Axis, Neuman Medical Ltd., Budapest, Hungary) containing 9090 previously performed CCTA datasets and structured reports.’
See metoprolol 20 mg IV to reach the goal? how often is it given? Is it supervised by physicians or RN.
The heart rate control routine followed in the current study is in accordance with the guidelines of the Society of Cardiovascular Computed Tomography (doi: 10.1016/j.jcct.2016.10.002). Although the guidelines state that up to 20-25 mg of intravenous metoprolol can be administered safely under the supervision of a physician, only one patient required four boluses of iv beta blocker. A detailed descriptive table of the administered doses of metoprolol has been included below and as supplementary material for the manuscript. All scans are performed under the supervision of a local physician at our institute who is responsible for the optimization of acquisition parameters and the administration of nitroglycerine and beta-blocker.
Table S1. Administered dose of metoprolol. |
||
|
|
|
Number |
160 |
|
Intravenous beta blocker (mg) |
|
|
0, n (%) |
73 (45.6) |
|
5, n (%) |
54 (33.8) |
|
10, n (%) |
24 (15.0) |
|
15, n (%) |
8 (5.0) |
|
20, n (%) |
1 (0.6) |
|
|
Moreover, the methods section was supplemented accordingly:
‘If the patient’s HR exceeded 65 beats/min, a maximum of 20 mg metoprolol was given intravenously for heart rate control under the supervision of a physician (Supplementary Table 1).’